# WHITENING FOR SELF-SUPERVISED REPRESENTATION LEARNING

## ABSTRACT

Most of the self-supervised representation learning methods are based on the *contrastive loss* and the instance-discrimination task, where augmented versions of the same image instance ("positives") are contrasted with instances extracted from other images ("negatives"). For the learning to be effective, a lot of negatives should be compared with a positive pair, which is computationally demanding. In this paper, we propose a different direction and a new loss function for self-supervised representation learning which is based on the *whitening* of the latent-space features. The whitening operation has a "scattering" effect on the batch samples, which compensates the use of negatives, avoiding degenerate solutions where all the sample representations collapse to a single point. Our Whitening MSE (W-MSE) loss does not require special heuristics (e.g. additional networks) and it is conceptually simple. Since negatives are not needed, we can extract multiple positive pairs from the same image instance. We empirically show that W-MSE is competitive with respect to popular, more complex self-supervised methods. The source code of the method and all the experiments is included in the Supplementary Material.

## 1 INTRODUCTION

One of the current main bottlenecks in deep network training is the dependence on large annotated training datasets, and this motivates the recent surge of interest in unsupervised methods. Specifically, in self-supervised representation learning, a network is (pre-)trained without any form of manual annotation, thus providing a means to extract information from unlabeled-data sources (e.g., text corpora, videos, images from the Internet, etc.). In self-supervision, label information is replaced by a prediction problem using some form of *context* or using a *pretext* task. Pioneering work in this direction was done in Natural Language Processing (NLP), in which the co-occurrence of words in a sentence is used to learn a language model (Mikolov et al., 2013a;b; Devlin et al., 2019). In Computer Vision, typical contexts or pretext tasks are based on: (1) the temporal consistency in videos (Wang & Gupta, 2015; Misra et al., 2016; Dwibedi et al., 2019), (2) the spatial order of patches in still images (Noroozi & Favaro, 2016; Misra & van der Maaten, 2019; Hénaff et al., 2019) or (3) simple image transformation techniques (Ji et al., 2019; He et al., 2019; Wu et al., 2018). The intuitive idea behind most of these methods is to collect pairs of *positive* and *negative* samples: two positive samples should share the same semantics, while negatives should be perceptually different. A triplet loss (Sohn, 2016; Schroff et al., 2015; Hermans et al., 2017; Wang & Gupta, 2015; Misra et al., 2016) can then be used to learn a metric space which should represent the human perceptual similarity. However, most of the recent studies use a contrastive loss (Hadsell et al., 2006) or one of its variants (Gutmann & Hyvärinen, 2010; van den Oord et al., 2018; Hjelm et al., 2019), while Tschannen et al. (2019) show the relation between the triplet loss and the contrastive loss.

It is worth noticing that the success of both kinds of losses is strongly affected by the number and the quality of the negative samples. For instance, in the case of the triplet loss, a common practice is to select *hard/semi-hard* negatives (Schroff et al., 2015; Hermans et al., 2017). On the other hand, Hjelm et al. (2019) have shown that the contrastive loss needs a large number of negatives to be competitive. This implies using batches with a large size, which is computationally demanding, especially with high-resolution images. In order to alleviate this problem, Wu et al. (2018) use a *memory bank* of negatives, which is composed of feature-vector representations of all the training samples. He et al. (2019) conjecture that the use of large and fixed-representation vocabularies is

one of the keys to the success of self-supervision in NLP. The solution proposed by He et al. (2019) extends Wu et al. (2018) using a memory-efficient queue of the last visited negatives, together with a *momentum encoder* which preserves the intra-queue representation consistency. Chen et al. (2020) have performed large-scale experiments confirming that a large number of negatives (and therefore a large batch size) is required for the contrastive loss to be efficient. Concurrently with our work, Grill et al. (2020) have suggested that it is not necessary to rely on the contrastive scheme, introducing a high-performing alternative based on bootstrapping.

In this paper we propose a new self-supervised loss function which first *scatters* all the sample representations in a spherical distribution[1] and then *penalizes* the positive pairs which are far from each other. In more detail, given a set of samples $V = \{\mathbf{v}_i\}$, corresponding to the current mini-batch of images $B = \{x_i\}$, we first project the elements of $V$ onto a spherical distribution using a *whitening* transform (Siarohin et al., 2019). The whitened representations $\{\mathbf{z}_i\}$, corresponding to $V$, are normalized and then used to compute a Mean Squared Error (MSE) loss which accumulates the error taking into account only positive pairs $(\mathbf{z}_i, \mathbf{z}_j)$. We do not need to *contrast* positives against negatives as in the contrastive loss or in the triplet loss because the optimization process leads to shrinking the distance between positive pairs and, indirectly, scatters the other samples to satisfy the overall spherical-distribution constraint.

In summary, our contributions are the following:

- We propose a new loss function, Whitening MSE (W-MSE), for self-supervised training. W-MSE constrains the batch samples to lie in a spherical distribution and it is an alternative to positive-negative instance contrasting methods.

- Our loss does not rely on negatives, thus including more positive samples in the batch can be beneficial; we indeed demonstrate that multiple positive pairs extracted from one image improve the performance.

- We empirically show that our W-MSE loss outperforms the commonly adopted contrastive loss when measured using different standard classification protocols. We show that W-MSE is competitive with respect to state-of-the-art self-supervised methods.

## 2 Background and Related Work

A typical self-supervised method is composed of two main components: a *pretext task*, which exploits some a-priori knowledge about the domain to automatically extract supervision from data, and a *loss function*. In this section we briefly review both aspects, and we additionally analyse the recent literature concerning feature whitening.

**Pretext Tasks.** The temporal consistency in a video provides an intuitive form of self-supervision: temporally-close frames usually contain a similar semantic content (Wang & Gupta, 2015; van den Oord et al., 2018). Misra et al. (2016) extended this idea using the relative temporal order of 3 frames, while Dwibedi et al. (2019) used a *temporal cycle consistency* for self-supervision, which is based on comparing two videos sharing the same semantics and computing inter-video frame-to-frame nearest neighbour assignments.

When dealing with still images, the most common pretext task is *instance discrimination* (Wu et al. (2018)): from a training image $x$, a composition of data-augmentation techniques are used to extract two different views of $x$ ($x_i$ and $x_j$). Commonly adopted transformations are: image cropping, rotation, color jittering, Sobel filtering, etc.. The learner is then required to discriminate $(x_i, x_j)$ from other views extracted from other samples (Wu et al., 2018; Ji et al., 2019; He et al., 2019; Chen et al., 2020).

Denoising auto-encoders (Vincent et al., 2008) add random noise to the input image and try to recover the original image. More sophisticated pretext tasks consist in predicting the spatial order of image patches (Noroozi & Favaro, 2016; Misra & van der Maaten, 2019) or in reconstructing large masked regions of the image (Pathak et al., 2016). Hjelm et al. (2019); Bachman et al. (2019) compare the holistic representation of an input image with a patch of the same image. Hénaff et al.

---

[1]Here and in the following, with "spherical distribution" we mean a distribution with a zero-mean and an identity-matrix covariance.

(2019) use a similar idea, where the comparison depends on the patch order: the appearance of a given patch should be predicted given the appearance of the patches which lie above it in the image.

In this paper we use standard data augmentation techniques on still images to obtain positive pairs, which is a simple method to get self-supervision (Chen et al., 2020) and does not require a pretext-task specific network architecture (Hjelm et al., 2019; Bachman et al., 2019; Hénaff et al., 2019).

**Loss functions.** Denoising auto-encoders use a *reconstruction loss* which compares the generated image with the input image before adding noise. Other generative methods use an *adversarial loss* in which a discriminator provides supervisory information to the generator (Donahue et al., 2017; Donahue & Simonyan, 2019).

Early self-supervised (deep) discriminative methods used a *triplet loss* (Wang & Gupta, 2015; Misra et al., 2016): given two *positive* images $x_i, x_j$ and a *negative* $x_k$ (Sec. 1), together with their corresponding latent-space representations $\mathbf{z}_i, \mathbf{z}_j, \mathbf{z}_k$, this loss penalizes those cases in which $\mathbf{z}_i$ and $\mathbf{z}_k$ are closer to each other than $\mathbf{z}_i$ and $\mathbf{z}_j$ plus a margin $m$:

$$L_{Triplet} = -\max(\mathbf{z}_i^T \mathbf{z}_k - \mathbf{z}_i^T \mathbf{z}_j + m, 0). \tag{1}$$

Most of the recent self-supervised discriminative methods are based on some *contrastive loss* (Hadsell et al., 2006) variant, in which $\mathbf{z}_i$ and $\mathbf{z}_j$ are contrasted against a set of negative pairs. Following the common formulation proposed by van den Oord et al. (2018):

$$L_{Contrastive} = -\log \frac{\exp\left(\mathbf{z}_i^T \mathbf{z}_j / \tau\right)}{\sum_{k=1, k \neq i}^{K} \exp\left(\mathbf{z}_i^T \mathbf{z}_k / \tau\right)}, \tag{2}$$

where $\tau$ is a *temperature* hyperparameter which should be manually set and the sum in the denominator is over a set of $K - 1$ negative samples. Usually $K$ is the size of the current batch, i.e., $K = 2N$, being $N$ the number of the positive pairs. However, as shown by Hjelm et al. (2019), the contrastive loss (2) requires a large number of negative samples to be competitive. Wu et al. (2018); He et al. (2019) use a set of negatives much larger than the current batch, by pre-computing latent-space representations of old samples. SimCLR (Chen et al. (2020)) uses a simpler, but computationally very demanding, solution based on large batches.

While recent works (van den Oord et al., 2018; Hénaff et al., 2019; Hjelm et al., 2019; Bachman et al., 2019; Ravanelli & Bengio, 2018) draw a relation between the contrastive loss and an estimate of the mutual information between the latent-space image representations, Tschannen et al. (2019) showed that the success of this loss is likely related to learning a metric space, similarly to what happens with a triplet loss. On the other hand, Wang & Isola (2020) showed that the $L_2$ normalized contrastive loss asymptotically converges to the minimization of two desirable characteristics of the latent-space representations on the surface of the unit hypersphere: uniformity and semantic alignment. In the same paper, the authors propose two new losses ($\mathcal{L}_{\text{uniform}}$ and $\mathcal{L}_{\text{align}}$) which explicitly deal with these characteristics.

Concurrently with our work, BYOL (Grill et al. (2020)) proposes a "bootstrapping" scheme which is alternative to the positive-negative contrastive learning. In BYOL, an "online" network is optimised to predict the output of a "target" network, whose parameters are a running average of the online network. The predictions of the two networks are compared using an additional prediction network and an MSE loss. However, very recently, Fetterman & Albrecht (2020) and Tian et al. (2020) have empirically shown that BYOL can avoid a collapsed solution through the use of the Batch Norm (BN) (Ioffe & Szegedy, 2015) which avoids constant representations. Our work can be seen as a generalization of this finding with a much simpler network architecture (more details in Sec. 3.1).

In this paper we propose a different loss which is competitive with respect to other alternatives. Our loss formulation is simpler because it does not require a proper setting of the $\tau$ hyperparameter in equation 2, $m$ in equation 1, or additional networks with a specific weight update schemes as in BYOL.

**Feature Whitening.** We adopt the efficient and stable Cholesky decomposition (Dereniowski & Marek, 2004) based *whitening* transform proposed by Siarohin et al. (2019) to project our latent-space vectors into a spherical distribution (more details in Sec. 3). Note that Huang et al. (2018);

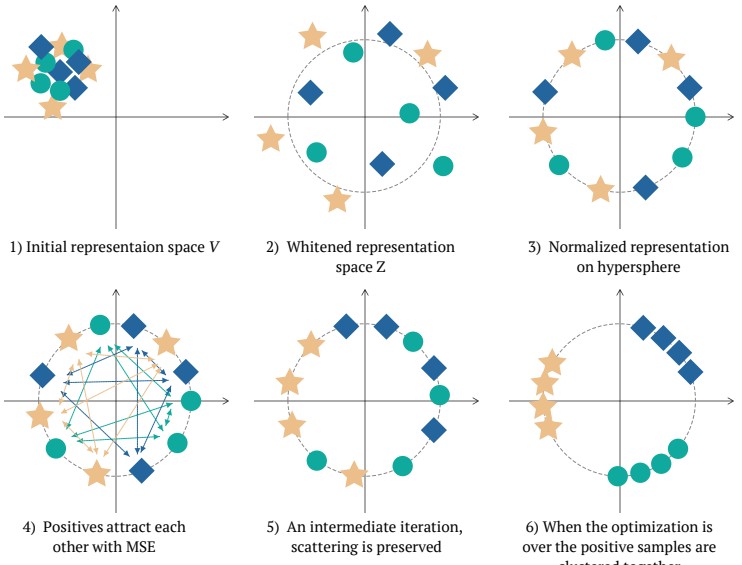

1) Initial representaion space $V$    2) Whitened representation space Z    3) Normalized representation on hypersphere

4) Positives attract each other with MSE    5) An intermediate iteration, scattering is preserved    6) When the optimization is over the positive samples are clustered together

Figure 1: A schematic representation of the W-MSE based optimization process. Positive pairs are indicated with the same shapes and colors. (1) A representation of the feature batch $V$ when training starts. (2, 3) The distribution of the elements after whitening and $L_2$ normalization. (4) The MSE computed over the normalized $\mathbf{z}$ features encourages the network to move the positive pair representations closer to each other. (5) The subsequent iterations move closer and closer the positive pairs, while the relative layout of the other samples is forced to lie in a spherical distribution.

Siarohin et al. (2019) use whitening transforms in the intermediate layers of the network for a completely different task: extending BN to a multivariate batch normalization.

## 3  THE WHITENING MSE LOSS

Given an image $x$, we extract an embedding $\mathbf{z} = f(x; \theta)$ using an encoder network $f(\cdot; \theta)$ parametrized with $\theta$ (more details below). We require that: (1) the image embeddings are drawn from a non-degenerate distribution (the latter being a distribution where, e.g., all the representations collapse to a single point), and (2) positive image pairs $(x_i, x_j)$, which share a similar semantics, should be clustered close to each other. We formulate this problem as follows:

$$min_\theta \, \mathbb{E} \, dist(\mathbf{z}_i, \mathbf{z}_j), \tag{3}$$
$$s.t. \, cov(\mathbf{z}_i, \mathbf{z}_i) = cov(\mathbf{z}_j, \mathbf{z}_j) = I, \tag{4}$$

where $dist(\cdot)$ is a distance between vectors, $I$ is the identity matrix and $(\mathbf{z}_i, \mathbf{z}_j)$ corresponds to a positive pair of images $(x_i, x_j)$. With equation 4, we constrain the distribution of the $\mathbf{z}$ values to be non-degenerate, hence avoiding that all the probability mass is concentrated in a single point. Moreover, equation 4 makes all the components of $\mathbf{z}$ to be linearly independent from each other, which encourages the different dimensions of $\mathbf{z}$ to represent different semantic content. We define the distance with the cosine similarity, implemented with MSE between normalized vectors:

$$dist(\mathbf{z}_i, \mathbf{z}_j) = \left\| \frac{\mathbf{z}_i}{\|\mathbf{z}_i\|_2} - \frac{\mathbf{z}_j}{\|\mathbf{z}_j\|_2} \right\|_2^2 = 2 - 2\frac{\langle \mathbf{z}_i, \mathbf{z}_j \rangle}{\|\mathbf{z}_i\|_2 \cdot \|\mathbf{z}_j\|_2} \tag{5}$$

In Appendix C we also include other experiments in which the cosine similarity is replaced by the Euclidean distance. We provide below the details on how positive image samples are collected, how they are encoded and how the above optimization is implemented.

First, similarly to Chen et al. (2020), we obtain positive samples sharing the same semantics from a single image $x$ and using standard image transformation techniques. Specifically, we use a composition of image cropping, grayscaling and color jittering transformations $T(\cdot; \mathbf{p})$. The parameters

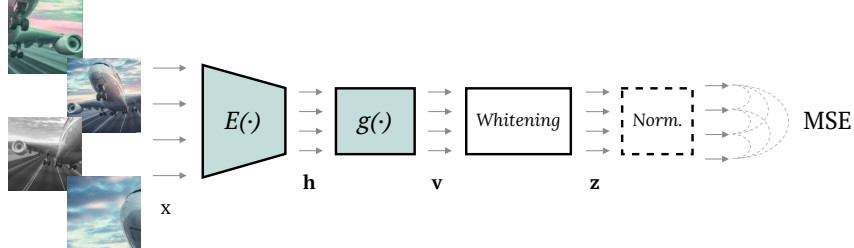

Figure 2: A scheme of our training procedure. First, $d$ ($d = 4$ in this case) positive samples are generated using augmentations. These images are transformed into vectors with the encoder $E(\cdot)$. Next, they are projected onto a lower dimensional space with a projection head $g(\cdot)$. Then, Whitening projects these vectors onto a spherical distribution, followed by an optional $L_2$ normalization. Finally, the dashed curves show all the $d(d-1)/2 = 6$ comparisons used in our W-MSE loss.

($\mathbf{p}$) are selected uniformly at random and independently for each *positive* sample extracted from the same image: $x_i = T(x; \mathbf{p}_i)$. We concisely indicate with $pos(i, j)$ the fact that $x_i$ and $x_j$ ($x_i, x_j \in B$, $B$ the current batch) have been extracted from the same image.

The number of positive samples per image $d$ may vary, trading off diversity in the batch and the amount of the training signal. Favoring more negatives, most of the methods use one positive pair ($d = 2$). However, Ji et al. (2019) have demonstrated improved performance with 5 samples, while Caron et al. (2020) use 8 samples. In our MSE-based loss (see below), we use all the possible $d(d-1)/2$ combinations of positive samples. We include experiments for $d = 2$ (1 positive pair) and $d = 4$ (6 positive pairs).

For representation learning, we use a backbone *encoder* network $E(\cdot)$. $E(\cdot)$, trained without human supervision, will be used in Sec. 4 for evaluation using standard protocols. We use a standard ResNet-18 (He et al., 2016) as the encoder, and $\mathbf{h} = E(x)$ is the output of the average-pooling layer. This choice has the advantage to be simple and easily reproducible, in contrast to other methods which use encoder architectures specific for a given pretext task (see Sec. 2). Since $\mathbf{h} \in \mathbb{R}^{512}$ is a high-dimensional vector, following Chen et al. (2020) we use a nonlinear projection head $g(\cdot)$ to project $\mathbf{h}$ in a lower dimensional space: $\mathbf{v} = g(\mathbf{h})$, where $g(\cdot)$ is implemented with a MLP with one hidden layer and a BN layer. The whole network $f(\cdot)$ is given by the composition of $g(\cdot)$ with $E(\cdot)$ (see Fig. 2).

Given $N$ original images and a batch of samples $B = \{x_1, ...x_K\}$, where $K = Nd$, let $V = \{\mathbf{v}_1, ...\mathbf{v}_K\}$, be the corresponding batch of features obtained as described above. In the proposed W-MSE loss we compute the MSE over all $Nd(d-1)/2$ positive pairs, where constraint 4 is satisfied using the reparameterization of the $\mathbf{v}$ variables with the whitened variables $\mathbf{z}$:

$$L_{W-MSE}(V) = \frac{2}{Nd(d-1)} \sum_{(\mathbf{v}_i, \mathbf{v}_j) \in V, pos(i,j)} dist(\mathbf{z}_i, \mathbf{z}_j), \qquad (6)$$

where $\mathbf{z} = Whitening(\mathbf{v})$, and:

$$Whitening(\mathbf{v}) = W_V(\mathbf{v} - \boldsymbol{\mu}_V). \qquad (7)$$

In equation 7, $\boldsymbol{\mu}_V$ is the mean of the elements in $V$: $\boldsymbol{\mu}_V = \frac{1}{K} \sum_k \mathbf{v}_k$, while the matrix $W_V$ is such that: $W_V^\top W_V = \Sigma_V^{-1}$, being $\Sigma_V$ the covariance matrix of $V$:

$$\Sigma_V = \frac{1}{K-1} \sum_k (\mathbf{v}_k - \boldsymbol{\mu}_V)(\mathbf{v}_k - \boldsymbol{\mu}_V)^T. \qquad (8)$$

For more details on how $W_V$ is computed, we refer to Appendix B. Equation 7 performs the full whitening of each $\mathbf{v}_i \in V$ and the resulting set of vectors $Z = \{\mathbf{z}_1, ..., \mathbf{z}_K\}$ lies in a zero-centered distribution with a covariance matrix equal to the identity matrix (Fig. 1).

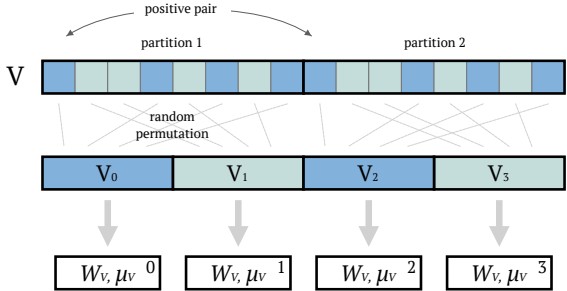

Figure 3: Batch slicing. $V$ is first partitioned in $d$ parts ($d = 2$ in this example). We randomly permute the first part and we apply the same permutation to the other $d - 1$ parts. Then, we further split all the partitions and we create sub-batches ($V_i$). Each $V_i$ is independently used to compute the sub-batch specific whitening matrix $W_V^i$ and centroid $\boldsymbol{\mu}_V^i$.

The intuition behind the proposed loss is that equation 6 penalizes positives which are far apart from each other, thus leading $g(E(\cdot))$ to shrink the inter-positive distances. On the other hand, since $Z$ must lie in a spherical distribution, the other samples should be "moved" and rearranged in order to satisfy constraint 4 (see Fig. 1).

**Batch Slicing.** The estimation of the Mean Square Error in equation 6 depends on the whitening matrix $W_V$, which may have a high variance over consecutive iteration batches $V_t, V_{t+1}, \dots$. For this reason, inspired by the resampling methods (Efron, 1982), given a batch $V$, we slice $V$ in different non-overlapping sub-batches and we compute a whitening matrix independently for each sub-batch. In more details, we first partition the batch in $d$ parts, being $d$ the number of positives extracted from one image. In this way, each partition contains elements extracted from different original images (i.e., no pair of positives is included in a single partition, see Fig. 3). Then, we randomly permute the elements of the each partition with the same permutation. Next, each partition is further split in sub-batches, using the heuristic that the size of each sub-batch ($V_i$) should be equal to the size of embedding ($\mathbf{v}$) times 2 (this prevents instability issues when computing the covariance matrices). Next, for each $V_i$, we use only its elements to compute a corresponding whitening matrix $W_V^i$, which is used to whiten the elements of $V_i$ only (Fig. 3). In the loss computation (equation 6), all the elements of all the sub-batches are used, thus implicitly alleviating the differences among the different whitening matrices. Finally, it is possible to repeat the whole operation several times and to average the result to get a more robust estimate of equation 6.

## 3.1 DISCUSSION

In a common *instance-discrimination* task (Sec. 2), e.g., solved using equation 2, the similarity of a positive pair ($\mathbf{z}_i^T \mathbf{z}_j$) is contrasted with the similarity computed with respect to all the other samples ($\mathbf{z}_k$) in the batch ($\mathbf{z}_i^T \mathbf{z}_k$, $1 \leq k \leq K, k \neq i$). However, $\mathbf{z}_k$ and $\mathbf{z}_i$, extracted from different image instances, can occasionally share the same semantics (e.g., $x_i$ and $x_k$ are two different image *instances* of the unknown "cat" *class*). Conversely, the proposed W-MSE loss does not force all the instance samples to lie far from each other, but it only imposes a soft constraint (equation 4), which avoids degenerate distributions.

Note that previous work (He et al., 2019; Hénaff et al., 2019; Chen et al., 2020) highlighted that BN may be harmful for learning semantically meaningful representations because the network can "cheat" and exploit the batch statistics in order to find a trivial solution to equation 2. However, our whitening transform (equation 7) is applied only to the very last layer of the network $f(\cdot)$ (see Fig. 2) and it is not used in the intermediate layers, which is instead the case of BN. Hence, our $f(\cdot)$ cannot learn to exploit subtle inter-sample dependencies introduced by batch-statistics because of the lack of other learnable layers on top of the $\mathbf{z}$ features.

Similarly to equation 6, in BYOL (Grill et al., 2020) an MSE loss is used to compare the latent representations of two positives computed by slightly different networks without contrasting positives with negatives (Sec. 2). However, the MSE loss alone is inclined to collapse the representations of all the images to a constant value, which would make the MSE computation equal to zero. In BYOL,

both the projection and the prediction sub-networks have BN layers, and, very recently, (Fetterman & Albrecht, 2020; Tian et al., 2020) have empirically shown that BYOL, without these BN layers, generates collapsed latent-space representations with a close-to-chance level classification accuracy. The reason of this behaviour seems to depend on the fact that the feature *standardization* in BN scatters the **z** values in a batch and avoids constant representations. Our W-MSE can be seen as a generalization of this implicit property of BYOL, in which the **z** values of the current batch are full-whitened, so preventing possible collapsing effects of the MSE loss. Importantly, we reach this result without the need of a target network or sophisticated training protocols.

Finally, note that using BN alone without whitening, as in W-MSE, and without additional networks, as in BYOL, is not sufficient. Indeed, if we just minimize an MSE after feature standardization, the network can easily find a solution where all the dimensions of the embedding represent the same feature. We have empirically verified this behaviour in preliminary experiments based on standardization, in which the network converges to a zero loss value after a few epochs but with a low classification accuracy.

## 4 EXPERIMENTS

We test our loss and its competitors on the following **datasets**.

- CIFAR-10 and CIFAR-100 (Krizhevsky & Hinton, 2009), two small-scale datasets composed of $32 \times 32$ images with 10 and 100 classes, respectively.

- Tiny ImageNet (Le & Yang, 2015), a reduced version of ImageNet, composed of 200 classes with images scaled down to $64 \times 64$. The total number of images is: 100K (training) and 10K (testing).

- STL-10 (Coates et al., 2011), also derived from ImageNet, with $96 \times 96$ resolution images. While CIFAR-10, CIFAR-100 and Tiny ImageNet are fully-labeled, STL-10 is composed of 5K labeled training samples (500 per class) and 100K unlabeled training examples from a similar but broader distribution of images. There are additional 8K labeled testing images.

- ImageNet-100, a random 100-class subset of ImageNet (the list of the 100 classes is published in (Wang & Isola, 2020)), consisting of unaltered ImageNet images.

**Setting.** The goal of our experiments is to compare W-MSE with state-of-the-art losses, isolating the effects of other settings, such as the architectural choices. For this reason, we use the same encoder $E(\cdot)$ ResNet-18 for all the experiments. We independently select the best hyperparameter values for every method and every dataset. Each method uses $L_2$ feature normalization unless otherwise stated. *Contrastive* refers to our implementation of the contrastive loss (equation 2) following the details in (Chen et al., 2020), with temperature $\tau = 0.5$. *BYOL* is our reproduction of (Grill et al., 2020), introduced concurrently with our work. For this method we use the exponential moving average with cosine increasing, starting from 0.99. *W-MSE 2* and *W-MSE 4* correspond to our method with $d = 2$ and $d = 4$ positives extracted per image, respectively. For CIFAR-10 and CIFAR-100, the slicing sub-batch size is 128, for Tiny ImageNet and STL-10, it is 256. For experiments W-MSE 2 for Tiny ImageNet and STL-10 we use 4 iterations of batch slicing, for all other experiments we use 1 iteration.

In all the experiments, we use the Adam optimizer (Kingma & Ba, 2014). For all the tested methods (including ours), we use the same number of epochs and the same learning rate schedule. Specifically, for CIFAR-10 and CIFAR-100, we use 1,000 epochs with learning rate $3 \times 10^{-3}$; for Tiny ImageNet, 1,000 epochs with learning rate $2 \times 10^{-3}$; for STL-10, 2,000 epochs with learning rate $2 \times 10^{-3}$. We use learning rate warm-up for the first 500 iterations of the optimizer, and a 0.2 learning rate drop 50 and 25 epochs before the end. We use a mini-batch size of $K = 1024$ samples. The dimension of the hidden layer of the projection head $g(\cdot)$ is 1024. The weight decay is $10^{-6}$. Finally, we use an embedding size of 64 for CIFAR-10 and CIFAR-100, and an embedding of size of 128 for STL-10 and Tiny ImageNet. For ImageNet-100 we use a configuration similar to the Tiny ImageNet experiments, and 240 epochs of training.

As a common practice when using ResNet-like architectures for small-size image resolutions, in all the experiments, except ImageNet-100, we have a first convolutional layer with kernel size 3,

Table 1: Classification accuracy (top 1) of a linear classifier and a 5-nearest neighbors classifier for different loss functions and datasets with a ResNet-18 encoder.

| Method | CIFAR-10 | | CIFAR-100 | | STL-10 | | Tiny ImageNet | |
|---|---|---|---|---|---|---|---|---|
| | linear | 5-nn | linear | 5-nn | linear | 5-nn | linear | 5-nn |
| Contrastive | 91.80 | 88.42 | 66.83 | 56.56 | 90.51 | 85.68 | 48.84 | 32.86 |
| BYOL | 91.73 | 89.45 | 66.60 | 56.82 | 91.99 | 88.64 | 51.00 | 36.24 |
| W-MSE 2 | 91.55 | 89.69 | 66.10 | 56.69 | 90.36 | 87.10 | 48.20 | 34.16 |
| W-MSE 4 | 91.99 | 89.87 | 67.64 | 56.45 | 91.75 | 88.59 | 49.22 | 35.44 |

Table 2: Classification accuracy on ImageNet-100. W-MSE (2 and 4) are based on a *ResNet-18* encoder. [†] indicates that the results are based on a *ResNet-50* encoder and the values are reported from (Wang & Isola, 2020).

| Method | linear (top 1) | linear (top 5) | 5-nn |
|---|---|---|---|
| MoCo [†] | 72.80 | 91.64 | - |
| $\mathcal{L}_{\text{align}}$ and $\mathcal{L}_{\text{uniform}}$ [†] | 74.60 | 92.74 | - |
| W-MSE 2 | 76.00 | 93.14 | 67.04 |
| W-MSE 4 | 79.02 | 94.46 | 71.32 |

stride 1 and padding 1. Additionally, in case of CIFAR-10 and CIFAR-100, we remove the first max pooling layer.

**Image Transformation Details.** We extract crops with a random size from 0.2 to 1.0 of the original area and a random aspect ratio from $3/4$ to $4/3$ of the original aspect ratio, which is a commonly used data-augmentation technique. We also apply horizontal mirroring with probability 0.5. Finally, we apply color jittering with configuration $(0.4, 0.4, 0.4, 0.1)$ with probability 0.8 and grayscaling with probability 0.1. For ImageNet-100 we follow details in (Chen et al., 2020): crop size from 0.08 to 1.0, stronger jittering $(0.8, 0.8, 0.8, 0.2)$, grayscaling probability 0.2, and Gaussian blurring with 0.5 probability.

**Evaluation Protocol.** The most common evaluation protocol for unsupervised feature learning is based on *freezing* the network encoder ($E(\cdot)$, in our case) after unsupervised pre-training, and then train a supervised *linear classifier* on top of it. Specifically, the linear classifier is a fully-connected layer followed by softmax, which is placed on top of $E(\cdot)$ after removing the projection head $g(\cdot)$. In all the experiments we train the linear classifier for 500 epochs using the Adam optimizer and the labeled training set of each specific dataset, without data augmentation. The learning rate is exponentially decayed from $10^{-2}$ to $10^{-6}$. The weight decay is $5 \times 10^{-6}$.

In our experiments, we also include the accuracy of a k-nearest neighbors classifier (k-nn, $k = 5$). The advantage of using this classifier is that it does not require additional parameters and training, and it is deterministic.

### 4.1 COMPARISON WITH THE STATE OF THE ART

Tab. 1 shows the results of the experiments on small and medium size datasets. For W-MSE, 4 samples are generally better than 2. The contrastive loss performs the worst in most cases. The W-MSE 4 accuracy is the best on CIFAR-10 and CIFAR-100, while BYOL leads on STL-10 and Tiny ImageNet, although the gap between the two methods is minor. In Appendix A, we plot the linear classification accuracy during training for the STL-10 dataset. The plot shows that W-MSE 4 and BYOL have a similar performance during most of the training. However, in the first 120 epochs, BYOL significantly underperforms W-MSE 4 (e.g., the accuracy after 20 epochs: W-MSE 4, 79.98%; BYOL, 73.24%), indicating that BYOL requires a "warmup" period. On the other hand, W-MSE performs well from the beginning. This property is useful in those domains which require a rapid adaptation of the encoder, e.g., due to the change of the data distribution in continual learning or in reinforcement learning.

Tab. 2 shows the results on a larger dataset (ImageNet-100). In that table, MoCo is the contrastive-loss based method proposed in (He et al., 2019), where a momentum encoder and a large queue of negatives are used to improve the contrast of the positive pairs with respect to the other samples (see Sec. 2). $\mathcal{L}_{\text{align}}$ and $\mathcal{L}_{\text{uniform}}$ are the two losses proposed in (Wang & Isola, 2020) (Sec. 2). Note that, while W-MSE (2 and 4) in Tab. 2 refer to our method with a *ResNet-18* encoder, the other results are reported from (Wang & Isola, 2020), where a much larger-capacity network (i.e., a *ResNet-50*) is used as the encoder. *Despite this large difference in the encoder capacity*, both versions of W-MSE significantly outperform the other two compared methods in this dataset.

## 4.2 TRAINING TIME COMPLEXITY

Following (Siarohin et al., 2019), the complexity of the whitening transform is $O(k^3 + Mk^2)$, where $k$ is the embedding dimension and $M$ is the size of the sub-batch used in the batch slicing process. Since $k < M$ (see Sec. 3), the whitening transform is $O(Mk^2)$, which is basically equivalent to the forward pass of $M$ activations in a fully-connected layer connecting two layers of $k$ neurons each. In fact, the training time is dominated by other architectural choices which are usually more computationally demanding than the loss computation. For instance, BYOL (Grill et al., 2020) needs 4 forward passes through 2 networks for each pair of positives. Hence, to evaluate the wall-clock time, we measure the time spent for one mini-batch iteration by all the methods compared in Tab. 1. We use the STL-10 dataset, a ResNet-18 encoder and a server with one Nvidia Titan Xp GPU. Time of one iteration: Contrastive - 459ms, BYOL - 602ms, W-MSE 2 - 478ms, W-MSE 4 - 493ms. The 19ms difference between Contrastive and W-MSE 2 is due to the whitening transform. Since the factual time is mostly related to the sample forward and backward passes, the $d(d-1)$ positive comparisons in equation 6 do not significantly increase the wall-clock time of W-MSE 4 with respect to W-MSE 2.

## 4.3 CONTRASTIVE LOSS WITH WHITENING

Table 3: Accuracy of the whitened contrastive loss on CIFAR-10 trained for 200 epochs.

| Method | linear | 5-nn |
|---|---|---|
| Contrastive | 89.66 | 86.55 |
| Contrastive with Whitening | diverged | |
| Contrastive, unnormalized features | 79.48 | 76.60 |
| Contrastive with Whitening, unnormalized features | 77.39 | 74.14 |

In this section, we analyse the effect of the whitening transform in combination with the contrastive loss. Tab. 3 shows the results. The first row refers to the standard contrastive loss. Note that the difference with respect to Tab.1 is due to the use of only 200 training epochs. The second row refers to equation 2, where the features ($\mathbf{z}$) are computed using equation 7 and then $L_2$ normalized, while in the last two rows, $\mathbf{z}$ is not normalized. If the features are whitened and then normalized, we observed an unstable training, with divergence after a few epochs. The unnormalized version with whitening converged, but its accuracy is worse than the standard contrastive loss (both normalized and unnormalized). This experiments show that whitening itself does not improve the performance, but it only allows to satisfy the constraint 4.

## 5 CONCLUSION

In this paper, we have proposed a new self-supervised representation learning loss, W-MSE, which is alternative to common loss functions used in the field. Differently from the triplet loss and the contrastive loss, both of which are based on comparing an instance-level similarity against other samples, W-MSE computes only the intra-positive distances, while using a whitening transform to avoid degenerate solutions. Despite W-MSE is very simple, its classification accuracy is comparable with state-of-the-art methods, achieving results significantly higher than MoCo, which requires an additional momentum encoder and a large queue of past samples. W-MSE is also comparable with BYOL, which needs an additional target network and a specific training protocol. We believe that the use of whitening to avoid collapsing effects can inspire other self-supervised methods.

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

## A  TRAINING DYNAMICS

Fig. 4 and 5 show the training dynamics for each of the considered losses. Charts are smoothed with a 0.3 moving average for readability (curves before smoothing are shown semi-transparent).

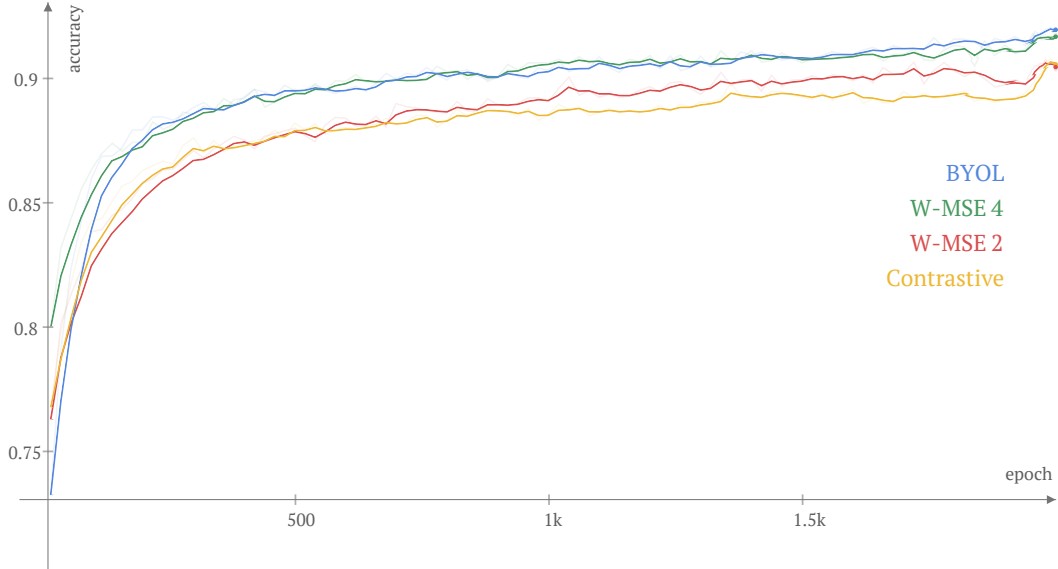

Figure 4: Training dynamics on STL-10 dataset for linear classifier

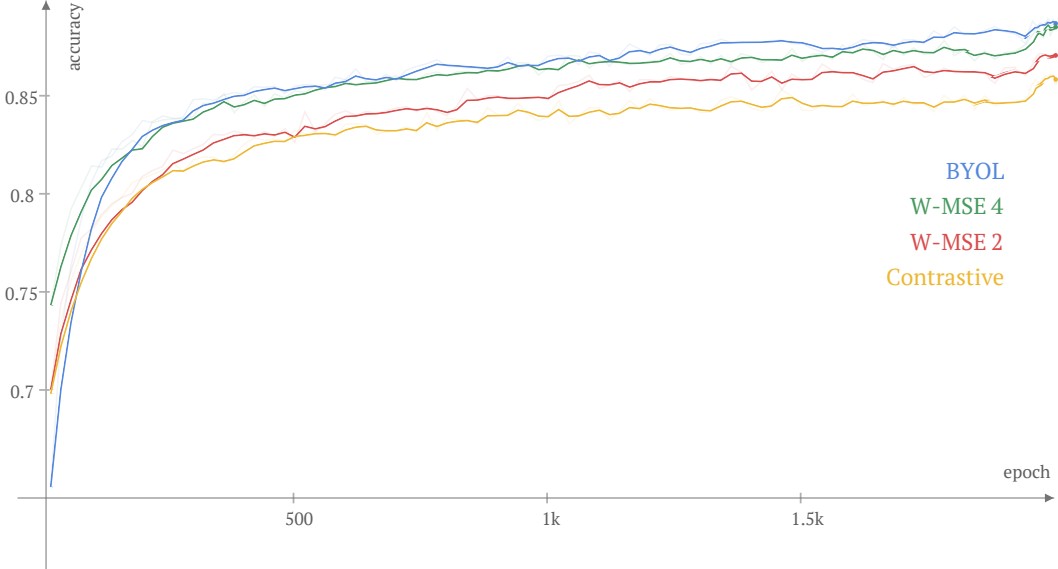

Figure 5: Training dynamics on STL-10 dataset for 5-nn classifier

## B  CHOLESKY WHITENING AND BACKPROGATION

We compute $W_V$ (equation 8) following (Siarohin et al., 2019) and using the Cholesky decomposition. The Cholesky decomposition is based on the factorisation of the covariance symmetric matrix using two triangular matrices: $\Sigma_V = LL^\top$, where $L$ is a lower triangular matrix. Once we get $L$, we compute the inverse of $L$, and we get: $W_V = L^{-1}$. Note that Cholesky decomposition is fully diferentiable and it is implemented in all of the major frameworks, such as PyTorch and TensorFlow. However, for the sake of completeness, we provide below the gradient computation.

## B.1 GRADIENT COMPUTATION

We provide here the equations for whitening differentiation. Let $Z$ be the whitened version of the batch $V$, i.e., $Z = W_V(V - \boldsymbol{\mu}_V)$ (equation 7). The gradient $\frac{\partial L}{\partial V}$ can be computed by:

$$\frac{\partial L}{\partial V} = \frac{2}{K-1}\frac{\partial L}{\partial \Sigma}V + W_V^T\frac{\partial L}{\partial Z}. \tag{9}$$

where the partial derivative $\frac{\partial L}{\partial Z}$ is backpropogated, while $\frac{\partial L}{\partial \Sigma}$ is computed as follows:

$$\frac{\partial L}{\partial \Sigma} = -\frac{1}{2}W_V^T\left(P \circ \frac{\partial L}{\partial W_V}W_V^T + \left(P \circ \frac{\partial L}{\partial W_V}W_V^T\right)^T\right)W_V \tag{10}$$

In equation 10, $\circ$ is Hadamard product, while $\frac{\partial L}{\partial W_V}$ is:

$$\frac{\partial L}{\partial W_V} = \frac{\partial L}{\partial Z}V^T, \tag{11}$$

and $P$ is:

$$P = \begin{pmatrix} \frac{1}{2} & 0 & \cdots & 0 \\ 1 & \frac{1}{2} & \ddots & 0 \\ 1 & \ddots & \ddots & 0 \\ 1 & \cdots & 1 & \frac{1}{2} \end{pmatrix}.$$

## C EUCLIDEAN DISTANCE

Table 4: Classification accuracy (top 1) using the Euclidean distance (unnormalized embeddings) on STL-10.

| Method | linear | 5-nn |
|---|---|---|
| Contrastive | 78.00 | 71.07 |
| BYOL | 80.83 | 74.94 |
| W-MSE 2 | 89.91 | 85.56 |
| W-MSE 4 | 90.40 | 87.09 |

The cosine similarity is a crucial component in most of the current self-supervised learning approaches. This is usually implemented with an $L_2$ normalization of the latent representations, which corresponds to projecting the features on the surface of the unit hypersphere. However, in our W-MSE, the whitening transform projects the representation onto a spherical distribution (intuitively, we can say on the whole unit hypersphere). Preserving the module of the features before the $L_2$ normalization may be useful in some applications, e.g., clustering the features after the projection head using a Gaussian mixture model. Tab. 4 shows an experiment on the STL-10 dataset where we use unnormalized embeddings for all the methods (and $\tau = 1$ for the contrastive loss). Comparing Tab. 4 with Tab. 1, the accuracy decrease of W-MSE is significantly smaller than the other methods.

