# OpenReview forum: "Whitening for Self-Supervised Representation Learning"
_ICLR.cc/2021/Conference — Reject_

### Official Review · AnonReviewer1 · 2020-10-26
**An interesting attempt to remove negative examples by whitening, however experiments are not too satisfying**

**Rating:** 7
**Confidence:** 4

**Review:**

The paper proposes to first do representation "whitening", so that the representations are scattered in the space and not collapsing to a single data point; then compute distance metric on top of that (e.g. Euclidean, cosine similarity). A nice thing about explicit scattering is that it does not require large numbers of negative examples to pull the features apart. Experiments are done on several toy datasets like CIFAR.

+ The paper is a nice, alternative attempt to remove negative examples in contrastive learning. Indeed, large number of negative examples is annoying and this direction is both exciting and significant.
+ The approach proposed in this paper intuitively makes sense. There are several works already explaining that contrastive learning is essentially doing some kind of scattering in the space.
+ The proposed approach seems pretty simple. The whitening code is only a dozen lines in PyTorch. I haven't run the code to verify the results though.

I think the experiments are not too satisfying though.

- The comparison is less fair between BYOL and multi-crop version of W-MSE. In SwAV, it shows that with multiple crops, the performance can be boosted quite a bit. I haven't tried on BYOL but I believe it could also be helping there. So the most apple-to-apple comparison between BYOL and W-MSE is the 2-crop version (d=2). In this case, BYOL is outperforming in most entries. Though it can be viewed as concurrent work (I think W-MSE is actually even earlier than BYOL), but the experiment session in the paper is not clear about this.
- Overall running experiments on these toy datasets are less satisfying, not only because it lacks comparison to other major approaches (like MoCo on ImageNet), but also because the signal we get from smaller datasets may not transfer well to more real-world images.
- I would like to see a comparison in terms of timing -- maybe BYOL (because of its momentum encoder and it needs 4 forward pass of the network to compute a single pair of losses) is running much smaller in training. W-MSE can run much faster because it only needs 4 (or even 1?) forward pass. This is a potential advantage that W-MSE has, but it is not clear from the paper.

Other than experiments, I am also not too satisfied with the writing. The paper mentions another paper when talking about the key method (how to do whitening, e.g. which is back propagated, which is not) when describing the central technique of the paper. I would like to see the paper more self-contained in the next version.

---

> ### Author Response · Authors · 2020-11-23
> **R1 reply**
>
> Q: multi-crop…
> A: Please, note that SwAV is another work concurrently proposed with respect to ours. Moreover, in SwAV, the multi-crop strategy is slightly different w.r.t. our proposed multiple positives (i.e., d > 2). Apart from the multiple resolutions of the crops in SwAV, another important difference is that we can compute d (d-1) positive differences in Eq. 6 with only d forward passes, while in SwAV, the number of comparisons grows linearly with d. Overall, we believe that the use of multiple positives is a (minor) contribution of our paper, and this is why we compare W-MSE 4 with BYOL. Apart from that, we agree that BYOL frequently (slightly) outperforms W-MSE. However, there are other advantages of W-MSE w.r.t. BYOL, such as the simplicity of the (explicit) mechanism we use to avoid a representation collapse (please, see the new discussion we added to Sec. 3.1 and the common answer to all the Reviewers).
>
> Q: toy datasets and comparison with MoCo
> A: We agree that larger-size datasets and comparison with other methods (e.g., MoCo) are important. For this reason, we used ImageNet-100 and we compared with the MoCo results reported in (Wang & Isola, 2020), obtaining a significant accuracy margin using a much smaller encoder (please, see the common answer to all the Reviewers).
>
> Q: timing…
> A: Please, see the common answer to all the Reviewers.
>
> Q: how to do whitening, e.g. which is back propagated, etc.
> A: For lack of space, we used the Appendix for this. Specifically, we added Appendix B which describes both the details of the whitening transform and the corresponding backpropagation procedure.

---

### Official Review · AnonReviewer2 · 2020-10-27
**Interesting method**

**Rating:** 6
**Confidence:** 3

**Review:**

##########################################################################

Summary:


The paper provides a simple self-supervision loss function which is computed using only positive samples. In particular, the proposed loss function is based on the whitening of a latent space-features. Results show competitive but not better results than other methods.

##########################################################################

Reasons for score:


Overall, I vote for accepting. The paper proposes a simple loss function based on whitening and the MSE loss. The method does not need negative samples to contrast with the positives since the whitening process shrinks the overall samples to satisfy the spherical-distribution while the MSE loss attracts the positive ones. Although I have some concerns with the validation I really like the method.


##########################################################################

Pros:


1. Interesting and simple method.

2. Clarity of the paper.


##########################################################################

Cons:

1. The validation

-- The evaluation has been carried with small and simple datasets (CIFAR,STL and Tiny ImageNet). BYOL authors presented an extensive experiment setup with several datasets. Why not use the same (at least some) databases and compare them with their reported results on the paper (and not use your own implemented version of BYOL)? In fact, some of those datasets are already used in this paper but as you use ResNet-18 instead of ResNet-50 (used in BYOL paper) the results are not comparable.

-- Why not ResNet-50 instead of ResNet-18. Validation would be easier with other published methods

-- W-MSE with d=4 seems to work better than d=2. What about larger d?

-- Although the authors point out that the proposed MSE loss does not need negative samples, negative samples are needed in the whitening transformation.

-- It would be great to decouple the system performance. What is the system performance using Withening with the classical contrastive loss? Which is the improvement provided by the MSE loss over the classical contrastive loss?

-- The authors say that Contrastive Loss needs large numbers of negative samples. How many negative samples were used in the experimental setup? The proposed method is evaluated with d=2 and d=4. Which is the positive/negative proportion of samples in the batches using the contrastive loss? How was this proportion selected?

Minor Comments:
- Please check: "On the other hand, Hjelm et al. (2019) have shown that the contrastive loss needs a large number of negatives to be competitive"



##########################################################################

Questions during the rebuttal period:


Please address and clarify the cons above


#########################################################################

Some typos:

(1) Page 4: matrix and (z_i,z_j) correspond -> corresponds
(2) Page 7: jitterering --> jittering


I think that the paper


UPDATE AFTER REBUTTAL:
The authors have covered most of my concerns about the paper and I think that the paper has been substantially improved. However, my biggest concern was about the experiment results and  I think the paper still lacks on validation comparison with other methods.

---

> ### Author Response · Authors · 2020-11-23
> **R2 reply**
>
> Q: evaluation…
> A: Please, see the answer (5) to R4 and the common answer to all the Reviewers.
>
> Q: ResNet-50 instead of ResNet-18
> A: ResNet-50 requires significantly more GPU memory and computational time, and, unfortunately, running these experiments in the short period of the rebuttal was not possible. Moreover, we could not find any ResNet-50 baseline for the small datasets. Please, note that in BYOL, the CIFAR experiments concern transfer learning from an ImageNet-pretrained encoder, while in our experiments in Tab. 1, all the encoders are trained from scratch on the corresponding datasets.
>
> Q: What about larger d?
> A: Please, see answer (3) to R4.
>
> Q: negative samples are needed in the whitening transformation
> A: We agree: the whitening transform is based on all the samples of the (sub-)batch and, indirectly, they are used as negatives. However, this operation is different from explicit contrasting, because the whitening transform uses the batch sample distribution to scatter the representations and avoid degenerate solutions. Please, see also the discussion in Sec. 3.1 and the common answer to all the Reviewers.
>
> Q: decouple the system performance
> A: We agree that this ablation is interesting, and we run a few experiments mixing the contrastive loss with the whitening transform. The results are discussed in the new Sec. 4.3.
>
> Q: Number of negative samples in the Contrastive Loss
> A: Concerning the contrastive loss, we strictly followed Chen et al. (2020) (SimCLR). For a given positive pair, all the other samples in the batch are negatives. Thus, the batch size controls this proportion. We use a batch size of 1024 for all the experiments.
>
> Q: check: "On the other hand, Hjelm et al. (2019)…
> A: This is mentioned in Sec. 3.1 of that paper and empirically verified in Sec. A. 3 of the same paper. Later, this observation was confirmed by other papers, e.g. (Chen et al. (2020)).
>
> Q: Typos
> A: We have corrected the typos and carefully proofread the whole paper, thank you for your suggestions.

---

### Official Review · AnonReviewer4 · 2020-10-28
**A simple and clean method, with very weak experiments.**

**Rating:** 5
**Confidence:** 4

**Review:**

This paper proposed an MSE loss function with whitening (W-MSE) for self-supervised representation learning. The motivation is to reduce the demand of negative examples in contrastive representation learning. The proposed W-MSE loss function is compared with popular contrastive loss on a few benchmarks.

Contrastive learning is a popular topic in the self-supervised learning domain. Most of the existing methods rely on negative samps to avoid trivial solutions. This paper proposed a simple and clean solution to tackle this problem, by using whitening in the loss term. This paper is also well explained and illustrated.

The main weakness of the paper is the experiments. Based on the results, I am not convinced that the proposed W-MSE is effective. Here are more comments:

(1) The experimental results are pretty close to the existing contrastive/BYOL baselines. On CIFAR-10 and STL-10, results are saturated thus the diff is very minor. On more challenging CIFAR-100 and Tiny ImageNet, the results are mixed when compared to BYOL.

(2) Given that the results are very close, what are the other benefits of using W-MSE loss? For example, is the training speed faster than the other methods (contrastive, BYOL) with negative sampling? It would be nice to include such results to demonstrate the effectiveness of W-MSE.

(3) This paper claims that without negative sampling, W-MSE loss encourages the use of more positive pairs in the batch. I'm wondering if the authors have tried more positive pairs beyond 4 in the paper.

(4) I cannot understand the motivation of ablating the popular methods with Euclidean distance. Cosine similarity is just simple and it costs nothing compared to Euclidean distance. I think ablating this from existing contrastive loss is not sufficient to show the effectiveness of BYOL.

(5) It would be nice to include a few comparable numbers from literature directly, instead of reproducing their methods. For example, this paper used ResNet18 while the BYOL paper used ResNet50. This paper reports results on Tiny ImageNet while the existing methods report on ImageNet. Note that I do not penalize this paper for this point, as computational cost is high for using larger architecture and larger dataset. But it would be good to have a comparable baseline in the middle ground, e.g. ResNet50 architecture on CIFAR-100 dataset, where BYOL reports 78.4% top-1 accuracy with linear eval.

======Post Rebuttal Update======

I would like to thank the authors for their rebuttal, which has addressed part of my concerns. After reading the authors' rebuttal and other reviewers' comments, I'm still concerned on the weak baselines and mixed results in this paper. Unfortunately, I will keep my rating.

Concrete suggestions to improve this paper in the future:

(1) Strongly recommended: use ResNet50 instead of ResNet18 for the small scale experiments, in this way you get directly the numbers from the literature (e.g. BYOL on CIFAR-100);

(2) Nice to have: for the expensive ImageNet experiments, it would be nice to get comparable results using the smallest comparable architecture (e.g. ResNet50) from the literature. SimCLR claims that "With 128 TPU v3 cores, it takes ∼1.5 hours to train our ResNet-50 with a batch size of 4096 for 100 epochs" and MoCo claims that "For IN-1M, we use a mini-batch size of 256 in 8 GPUs, ..., train for 200 epochs ..., taking ∼53 hours training ResNet-50."

---

> ### Author Response · Authors · 2020-11-23
> **R4 reply**
>
> (1, 2, 4) Please, see the common answer to all the Reviewers.
>
> (3) In our experiments with a fixed batch size, d = 5 positive samples perform similarly to d = 4, while, with a larger number of positives, the performance starts to decrease. Note that the value of d is related with the sub-batch size, and it cannot grow too much to avoid instability issues (please, see the batch slicing process in Sec. 3 and the new details we added to that paragraph).
>
> (5) We agree that using results reported in the literature makes our method more easily comparable with other approaches. We used the results reported in (Wang & Isola, 2020) for the new experiment with ImageNet-100 (please, see also the common answer to all the Reviewers).

---

### Official Review · AnonReviewer3 · 2020-10-29
**A well-established method for unsupervised learning with contrastive loss functions with not very impressive results.**

**Rating:** 5
**Confidence:** 4

**Review:**

This paper proposes the mean square error loss with whitening operation to project positive pairs closely to each others while projecting the different positive pairs far away from each other on a unit sphere. This way, similar to BYOL, this paper removes the construction of negative pairs while improving the MoCo-V2 slightly on not very challenging benchmarks. The authors can find my questions/comments in the list below.

1. What does gray and blue squares represent in figure 3? I believe figure 3 can be revisited for the sake of better understanding the method. For example, I was not able to understand why v_i and v_i+8 represent the same image. In this current shape, it creates confusion rather than helping for clarity.

2. In the Batch Slicing section, the authors mention that the size each of sub-batch should be close to the size of the embedding x 2. Does this make sense? Is the size of embedding 512? If yes, do you have 1024 samples in a sub-batch? I would be happy if the authors can comment on this.

3. I think my biggest concern with the paper is the lack of extensive experiments. All the experiments are done on small-scale datasets which is highly questionable when they are used for unsupervised learning. It would be much more convincing to have experiments on ImageNet which is a standard experiment for unsupervised learning.

4. The improvement on relatively less challenging benchmarks are very marginal. Even in some cases, CIFAR100, contrastive loss does better than the proposed one. And the proposed method only outperforms BYOL in CIFAR10.

5. My impression is that the contrastive method in table 1 and 2 represent the MoCo-v2. I would replace it with MoCo-v2 in these tables as there are many other methods that uses contrastive loss functions.

6. Another problem with the experiments is that, they lack experiments on different downstream tasks such as object detection.

7. The authors try the Euclidean and MSE distance to project positives closely. Did they consider the cosine similarity loss? I did not understand the point of using Euclidean distance experiments without normalization? What do they exactly prove? And we can always normalize the embeddings as it is done in the other methods. I would be happy to receive some comments on this from the authors.

8. Finally, it would be nice to the advantages of the proposed method (removal of negatives) in terms of training complexity. Does it, as expected, reduce the complexity in the training time? If yes, quantifying it would increase the strength of the paper.

---

> ### Author Response · Authors · 2020-11-23
> **R3 reply**
>
> (1) We agree that the figure was not clear enough. We have changed that figure and added more details to the batch slicing process in Sec. 3.
>
> (2) We use a sub-batch size equal to the embedding dimension times 2 to avoid instability issues when computing the covariance matrices. We added more details on this and on the whole slicing process in Sec. 3.
>
> (3, 4) Please, see the common answer to all the Reviewers.
>
> (5) Actually, “Contrastive” in the paper refers to (our reproduction of) SimCLR (Chen et al. (2020)). In the new version of the manuscript, we compare with MoCo (version 1) in the new Tab. 2.
>
> (6) Most of the transfer learning-based experiments in BYOL (and in other papers as well) are based on fine-tuning an encoder pretrained on ImageNet.  Unfortunately, ImageNet training is very computationally demanding (please, see also the common answer to all the Reviewers). However, we believe that the simplicity of our W-MSE loss is an advantage which makes it possible to easily reproduce our method and possibly plug and test it in other self-supervised approaches.
>
> (7, 8) Please, see the common answer to all the Reviewers.

---

### Author Response · Authors · 2020-11-23
**Revision and general answers**

We thank all the reviewers for their constructive feedback and their comments. Most of the reviewers have appreciated the originality of the proposed method, although they highlight that the experiments are based on relatively small datasets and that the accuracy gain of our proposal with respect to BYOL and the contrastive loss is not always clear. Unfortunately, large-scale ImageNet experiments require a strong engineering effort and a huge computational budget. E.g., Chen et al. (2020) (SimCLR, i.e., our reference for the contrastive loss) report using from 32 to 128 TPU v3 cores, while Grill et al. (2020) (BYOL) report using 512 TPU v3 cores. However, we agree that larger-scale datasets are necessary to validate the proposed method. For this reason, in the new version of the manuscript we show a new experiment on ImageNet-100 and we compare with the results reported in (Wang & Isola, 2020), which include the well-known MoCo method (He et al., 2019). The new Tab. 2 shows that our W-MSE is significantly better than the other competitors tested on this dataset. Importantly, the accuracy gain was obtained using a ResNet-18, a much lower capacity encoder than the ResNet-50 used by the other approaches reported in (Wang & Isola, 2020). We believe that this result is very significant, and it will be included in the final version of the paper.

Concerning the comparison with BYOL, we acknowledge that our method is basically on par with (Grill et al. (2020)), being the accuracy differences in our experiments usually quite small and sometimes in favour of BYOL. However, BYOL is a concurrent approach, not published yet at the ICLR submission time. More importantly, recent works (Fetterman & Albrecht (2020) and (Tian et al. (2020)) have empirically showed that a representation collapse in BYOL is avoided because of the use of the Batch Norm (BN) in the projection and in the prediction sub-networks. Basically, the target and the online networks in BYOL converge toward a constant uninformative image representation when BN is removed. This very recent finding, which was not clear in (Grill et al. (2020)), shows that the BN feature standardization has the implicitly side-effect to scatter the feature values before computing the MSE in BYOL. On the other hand, our Whitening transform makes this mechanism explicit (please, see the constraint in Eq. 4) and more general. Importantly, we reach a similar effect without the use of an ad hoc architecture (i.e., the BYOL target network, etc.). For these reasons, we believe that, although the empirical results of W-MSE and BYOL are comparable with each other, the interest of our paper is to provide an explicit mechanism to avoid collapsed representations which is very simple and more general than BYOL. We added a discussion about this in Sec. 3.1.

Another common question is the computational time of W-MSE and its advantage w.r.t. BYOL and SimCLR. We added this analysis in Sec. 4.2, where we show that the loss computation time (e.g., in W-MSE) is negligible w.r.t. other methodological aspects (e.g., in BYOL, for each pair of positives, 4 forward passes through 2 networks need to be computed).

Finally, two Reviewers asked the motivation behind the Euclidean distance-based comparison. We believe that comparing the accuracy of different losses without an L2 normalization of the features is interesting because it shows that our approach is less sensitive w.r.t. this widely adopted normalization. However, we agree that this experiment is less important and we moved the corresponding analysis to the Appendix, while including in the new paper the aforementioned ImageNet-100 results and another experiment, in which Whitening is used in conjunction with the contrastive loss  (please, see answer 5 to R2).

---

### Decision · Program_Chairs · 2021-01-07
**Final Decision**

**Decision:**

Reject

**Comment:**

This paper received borderline recommendations (5, 5, 6, 7) but even the two slightly more positive reviewers were lukewarm (R1 and R2). While the reviewers acknowledged the heavy computational requirements to do an apples-to-apples comparison with existing baselines, they remain underwhelmed with the lack of experiments. I agree with their criticism; even though the proposed idea seems promising, without comprehensive experiments, it is difficult to judge the significance of this work. R1 commented after the discussion period that an earlier version of this paper actually had ImageNet results. R4 made excellent suggestions to improve the paper further. The authors are strongly encouraged to incorporate them into their future submission.

(I am copying R1's comment below in case it is invisible to the authors after the notification.)

Sorry for the late update -- I have read the rebuttal earlier. I would like to keep my acceptance rating but after the rebuttal I am fine either way. The paper first appeared in March on ArXiv, so indeed it is a concurrent work (actually an earlier work compared to BYOL or SwAV). We have actually tried to reproduce the results in the paper a while back but it did not go well (could not reproduce it), but this time the submission also includes the code. While I haven't run it, I trust the results are reproducible (maybe there are some tricks that I am not aware of).

Regarding running experiments on toy examples -- I can understand that this research is resource-constrained for ImageNet, but the earlier draft actually had some results on ImageNet (60+ top-1 accuracy) (see appendix of https://arxiv.org/pdf/2007.06346v1.pdf), and for some reason this submission removed that. So this is not a positive sign. Overall, my experience for CIFAR vs ImageNet is that it is easier to make things work on CIFAR, while it is much harder to do so on ImageNet. So maybe some trials are indeed done by the authors, but they choose to not report it in the submission for some reason. On the other hand, one can argue that results on toy datasets are good enough contributions for an early develop of something and they are just not ready for larger and more challenging datasets yet.

Therefore, this paper is quite a struggle. I hoped to see a better-than-this submission as this paper actually had all the time from March to October to improve its quality of experiments (actually even for ImageNet, one can to dozens of cycles on it during this time), but it did not for some reason.